# Systematic Review and Meta-Analysis of the Occurrence of ESKAPE Bacteria Group in Dogs, and the Related Zoonotic Risk in Animal-Assisted Therapy, and in Animal-Assisted Activity in the Health Context

**DOI:** 10.3390/ijerph17093278

**Published:** 2020-05-08

**Authors:** Antonio Santaniello, Mario Sansone, Alessandro Fioretti, Lucia Francesca Menna

**Affiliations:** 1Departments of Veterinary Medicine and Animal Productions, Federico II University of Naples, 80134 Naples, Italy; alessandro.fioretti@unina.it (A.F.); menna@unina.it (L.F.M.); 2Department of Electrical Engineering and Information Technology, Federico II University of Naples, 80125 Naples, Italy; mario.sansone@unina.it

**Keywords:** animal assisted interventions, zoonoses, dogs, one health, ESKAPE

## Abstract

Animal-assisted interventions are widely implemented in different contexts worldwide. Particularly, animal-assisted therapies and animal-assisted activities are often implemented in hospitals, rehabilitation centers, and other health facilities. These interventions bring several benefits to patients but can also expose them to the risk of infection with potentially zoonotic agents. The dog is the main animal species involved used in these interventions. Therefore, we aimed at collecting data regarding the occurrence of the pathogens ESKAPE (*Enterococcus faecium*, *Staphylococcus aureus*, *Klebsiella pneumoniae*, *Acinetobacter baumannii*, *Pseudomonas aeruginosa*, *Enterobacter* spp.) in dogs, in order to draft guidelines concerning the possible monitoring of dogs involved in animal-assisted therapies and animal-assisted activities in healthcare facilities. We performed a literature search using the PRISMA guidelines to examine three databases: PubMed, Web of Science, and Scopus. Out of 2604 records found, 52 papers were identified as eligible for inclusion in the review/meta-analysis. Sixteen papers reported data on *E. faecium*; 16 on *S. aureus*; nine on *K. pneumoniae*; four on *A. baumannii*; eight on *P. aeruginosa*; and six on *Enterobacter* spp. This work will contribute to increased awareness to the potential zoonotic risks posed by the involvement of dogs in animal-assisted therapies, and animal-assisted activities in healthcare facilities.

## 1. Introduction

Animal Assisted Interventions (AAIs) comprise a broad array of planned activities that involve animals for the purpose of improving human health and wellness. AAIs involve activities with teams of humans and animals with the aim of achieving therapeutic (Animal Assisted Therapy; AAT) or educational (Animal Assisted Education; AAE) goals. The AAIs also include Animal-Assisted Activities (AAA), such as informal interactions/visitations conducted on a volunteer basis by those teams for motivational, educational, or recreational purposes [1]. AATs provide well-being, promote the health of the patients [2], and assist in cognitive, emotional–affective, social, and linguistic rehabilitation [3,4,5,6].

In particular, results of several studies mainly involving dogs indicate significant benefits of AAT in people with psychophysical and mental health disorders, such as adults with Autism Spectrum Disorder [7], as well as Alzheimer disease and other dementias [6,8,9], and also during psychotherapy for adolescents [10]. As reported by Serpell and colleagues [11], various animal species are used in AAIs, but the dog is the most widely used species, especially in the AAT and in the AAA [3,12,13]. In AAT and AAA, patients interact with dogs by inter-specific relationship activities involving as petting, physical contact, brushing, playing, and strolling with the dog. Shen et al. [12] demonstrated that bodily contact may contribute significantly to AAT effectiveness. On the other hand, during these activities, the patients (often very young or old, or immunocompromised) may have physical contact with the dog’s mucosae and fur, and can be exposed to the bacteria, fungi, and parasites sub-clinically carried by the dog [14,15,16]. Therefore, a conflict exists between the need to preserve bodily contact during AAT and AAA, while reducing the risk of transmission of zoonotic agents. 

As reported in scientific literature, different bacterial species can be carried by the dog and transmitted to humans [17,18,19,20,21,22].

ESKAPE bacteria (i.e., *Enterococcus faecium*, *Staphylococcus aureus*, *Klebsiella pneumoniae*, *Acinetobacter baumannii*, *Pseudomonas aeruginosa*, and *Enterobacter* species) are a group of common opportunistic pathogens associated mainly with nosocomial infections [23]. The acronym ESKAPE was used for the first time in 2008 by Rice [23] and was coined to reflect these microorganism’s ability to escape killing by antibiotics by developing antimicrobial resistance, and challenge eradication by conventional therapies. The ESKAPE group of bacteria cause significant morbidity and mortality and increased resource utilization in healthcare facilities [24,25]. Moreover, World Health Organization (WHO) has recently listed most ESKAPE bacteria in the list of 12 microorganism against which new antibiotics are urgently needed [26].

*Enterococcus faecium* is commensal microorganism of the normal gastrointestinal flora in humans and animals. *E. faecium* can be transmitted to humans via direct contact with livestock as well as companion animals [27]. Recently, the results of some studies highlighted the potential for zoonotic transmission of ampicillin- and vancomycin-resistant *E. faecium* from the dog [27,28,29,30].

*Staphylococcus aureus* is part of the cutaneous microbiome of animals and humans and is one of the leading causes of fatal nosocomial infection in humans [31]. It can cause a range of infections, such as mild-to severe skin and soft tissue infections, endocarditis, osteomyelitis, and fatal pneumonia [32]. According to the sensitivity to antibiotic drugs, *S. aureus* can be divided into methicillin-sensitive *Staphylococcus aureus* (MSSA) and methicillin-resistant *Staphylococcus aureus* (MRSA). MRSA is one of the most significant bacteria causing both hospital and community-acquired infections in humans [17].

*K. pneumoniae* is Gram-negative member of the Enterobacteriaceae, considered one of the common opportunistic agents causing respiratory and urinary tract infections in humans and dogs [33,34,35,36,37]. *K. pneumoniae* strains have a significant ability to acquire resistance to antibiotics, and as such, it is of a public health concern. [38]. Marques and colleagues [35] reported the fecal colonization and sharing of *K. pneumoniae* clonal lineages between healthy humans and dogs living in close contact, suggesting the role of dogs as reservoirs of this bacterium.

*A. baumannii* is the most clinically significant pathogen implicated in human nosocomial infections [39]. In humans, *A. baumannii* infections involve mainly the respiratory tract, but meningitis and urinary tract infections may also occur [40,41,42]. Animals represent a potential reservoir of *A. baumannii,* and the risk of transmission could increase in companion animals which are in direct contact or closer vicinity to humans [43].

*P. aeruginosa* is increasingly recognized as an opportunistic pathogen causing chronic and recurrent infections in both humans and animals [44]. In humans, it causes nosocomial and healthcare-associated infections in immunocompromised patients [45,46]. Fernandes and colleagues [47] demonstrated a zoo-anthroponotic transmission (human–to-dog) of VIM-2–producing *P. aeruginosa* in the household following a person’s hospital discharge. 

*Enterobacter* spp., particularly *E. aerogenes* and *E. cloacae*, have been associated with nosocomial foci and are considered opportunistic pathogens [48]. *Enterobacter* spp. can cause numerous types of infections, including brain abscess, pneumonia, meningitis, septicemia, urinary tract (especially catheter-related) infections, and intestinal infections [49]. Transmission occurs through direct or indirect contact of the mucosal surfaces with the host organism [50].

There are few reports concerning the risks related to infections with ESKAPE bacteria in the contexts of AAIs, particularly in the AAT, and in the AAA in healthcare facilities. Therefore, the present review and meta-analysis aims at analyzing the published data on the presence of ESKAPE bacteria in the dog, in order to assess the risk of zoonotic transmission in these contexts. By our study, we intend to make an indirect assessment of the zoonotic risk deriving from contact with the dog, assuming its role as a healthy and/or asymptomatic carrier of ESKAPE bacteria in AAIs, but in particular in AAT context and in the AAA in healthcare facilities.

## 2. Materials and Methods 

### 2.1. Framework for Systematic Review of the Literature

This Review and Meta-Analysis was carried out applying the steps established by the PRISMA (Preferred Reporting Items for Systematic reviews and Meta-Analyses) group [51] as follows: (1), conduct a database search to obtain potentially pertinent articles, (2) assess the pertinency of papers (3), evaluate their quality, and (4) extract the data. The search strategy and article screening process are summarized in Figure 1.

### 2.2. Literature Search Strategy and Data Collection

The systematic literature search was performed using the following word strings: Enterococcus faecium AND “dog”, Staphylococcus aureus AND “dog”, Klebsiella pneumoniae AND “dog”, Acinetobacter baumannii AND “dog”, Pseudomonas aeruginosa AND “dog”, *Enterobacter* spp. AND “dog”. The extracted articles were sorted by the title and abstract and examined to remove duplicate and irrelevant articles. Only articles reporting original research in English (published or in press) were included, while reviews, case reports, retrospective analysis studies, comments, letters, etc., without reporting original data were excluded. Only articles published between the years 2000 and 2020 were used. Three scientific electronic databases were used: PubMed [52], Scopus [53], and Web of Science [54].

### 2.3. Relevancy Assessment of the Articles

The initially selected articles were classified as eligible for full text review if containing information about the occurrence of bacteria ESKAPE by dogs; whereas those focusing on the epidemiology of these bacteria only in humans or in healthcare facilities were excluded. No restrictions were applied regarding age, breed, health status, and living conditions of dogs, nor to the technique used to detect and identified bacteria ESKAPE.

The full texts of the recovered articles were examined for eventual inclusion. Papers were selected if they met the following inclusion criteria:Any paper published between January 2000 and January 2020 indicating the presence of one or more ESKAPE bacteria in dogs.Any paper that reported clear details about the type of the samples (swabs, biological specimens, medical instruments, such as intravenous catheter) and experimental design (number of dogs, percentage of positive dogs, dog category, health status of dog, geographical area, etc.).Any paper reporting the presence of one or more ESKAPE bacteria both in dogs and in humans (data on human samples were not considered).

Regarding the studies on the presence of *Staphylococcus aureus*, we have considered only those concerning MRSA, since the international scientific literature almost exclusively returned papers indexed on this particular topic.

Studies regarding the biological characteristics (i.e., phenotypic and molecular characterization of antimicrobial resistance) of strains of ESKAPE bacteria previously isolated from dogs were excluded. Furthermore, studies that referred to the presence of one or more ESKAPE species reporting negative results (no positive samples) were also excluded. Likewise, all studies published before 2000 were excluded.

### 2.4. Quality Evaluation and Data Extraction

Two researchers performed independently the full texts analysis of each record, using a data extraction form in order to obtain predetermined and methodological clear information qualitative and quantitative data; inconsistencies anyhow were decided by consensus. Data that consisted of first author/year of publication, number/type of sampled dogs, dog category, health status of dog, number of positives, and country and continent were extracted from included eligible articles. All data were insert in an Excel dataset. The independent researchers examined eligibility of studies according the criteria reported above, excluding they if there was not methodological enough information.

### 2.5. Meta-Analysis

Meta-analysis is a method to obtain a weighted average of results from various studies. In this manuscript we used methods for the meta-analysis of prevalence [55]. As the studies available have been conducted in different countries and using different kind of sampling procedures, we considered random-effect models in order to better account for this variability [56]. Moreover, heterogeneity of studies has been evaluated using I^2^ [57]. Heterogeneity was considered high for I^2^ > 50%. With the aim to investigate the origin of high heterogeneity, according to literature [58,59], we performed a subgroup analysis using “health status” as grouping variable. Further, in order to appreciate weights for single studies, we used the inverse variance method with arcsin transformation [60]. The meta-analysis has been conducted in R (Foundation for Statistical Computing, Vienna, Austria) [61] using the package metaphor [62]. Forest plots give a commonly used graphical summary of meta-analysis results [63]: they have been produced to easily visualize studies and averages, and they have been made available as supplementary material downloadable from the journal website. 

## 3. Results

The preliminary database search returned 2604 reports. Removal of duplicates yielded 1562 single papers. Each report was considered duplicated when it had the same information regarding the author, year of publication, name of the peer review, volume issue, and number of pages. All papers that did not meet the inclusion criteria were excluded, while 52 papers were selected for data extraction and qualitative analysis. Our results have been organized in seven tables. In the Table 1, most relevant pathogens based on number of included studies and the different geographical areas were reported, while in the other Table 2, Table 3, Table 4, Table 5, Table 6, Table 7, details of included studies per each ESKAPE bacterium were reported, respectively. Table 2, Table 3, Table 4, Table 5, Table 6, Table 7, include respectively sixteen articles reporting on *Enterococcus faecium* [27,28,29,30,64,65,66,67,68,69,70,71,72,73,74,75]; sixteen reporting on *Staphylococcus aureus* (methicillin resistant) [76,77,78,79,80,81,82,83,84,85,86,87,88,89,90,91]; nine reporting on *Klebsiella pneumoniae* [34,92,93,94,95,96,97,98,99]; four on *Acinetobacter baumannii* [100,101,102,103]; eight on *Pseudomonas aeruginosa* [45,82,94,98,104,105,106,107]; and six studies reporting on *Enterobacter* spp. [82,95,98,99,108,109]. The authors point out that some articles [82,94,95,98,99] provided data on more than one ESKAPE bacterial species, therefore, the same papers can be found in different tables. In each table, the articles have been placed in descending order, starting with the most recent.

### 3.1. Enterococcus faecium

In our study, a total of 2700 dogs were sampled in the 16 selected papers regarding *E. faecium* presence. The number of sampled dogs ranged from a minimum of 32 [67] to a maximum of 479 [75], while the number of positive dogs was from 3/32 (9.4%) [67] to 124/155 (80.0%) [71]. The type of samples taken for the isolation of *E. faecium* in the articles included in this study consisted of feces [27,28,29,30,68,70,71,73,74], urines [64,75], and other types of samples such as rectal [65,66], oral [67], several [69,72] swabs. Most of the articles (13 out of 16) were carried out on owned dogs [28,29,30,64,65,66,67,68,69,71,72,73,75], two studies did not show the dog category [70,74], while only one study carried out the investigation on military working dogs [27]. Of a total of 16 articles found, 6 were performed on healthy dogs [27,30,66,71,72,73], 3 on healthy and sick dogs [29,64,65], 4 on sick or hospitalized dogs [67,68,69,75], while 3 studies showed no indication regarding the health status of the sampled dogs [28,70,74]. For all details see Table 1.

Meta-Analysis results suggested that the overall prevalence is 0.30 (Confidence Interval (C.I.) 0.17–0.44). Study heterogeneity was I^2^ = 97%; consequently, we attempted to group the studies according to “health status of dog” and heterogeneity lowered to I^2^ = 52% (*p* < 0.01). All details regarding the meta-analysis results has been showed in Figure 2. 

### 3.2. Staphylococcus aureus

In the 16 articles included in this study and published from 2007 to 2019, a total of 2228 dogs were sampled. The positivity of dogs for this bacterium was detected in the following two main categories of samples: nasal and perineum swabs and nasal samples; in the first, it ranged between 1 (0.5%) and 25 (5.7%); in the second, between 1 (1.42%) and 8 (12.7%); in the other, samples were ocular swabs, pyogenic lesions, and generically swabs. In particular, the highest positivity of 8/16 (50.0%) in pyogenic lesions was reported by Ekapopphan and colleagues [82]; instead, the lowest positivity was reported by Hanselman and colleagues [90] in nasal and rectal swabs. Out of the searched articles on MRSA presence in dogs, 13 papers were carried out on owned dogs [76,77,79,80,81,82,84,86,87,88,89,90,91], one study was conducted on sheltered dogs [78], Tarazi and colleagues [85] sampled owned, stray and farm dogs, while only one study showed no indication regarding the health status of the sampled dogs [83]. The study conducted by Ekapopphan and colleagues [82] included also the evaluation of *P. aeruginosa* and *Enterobacter* spp. presence. All info regarding the articles about MRSA were included in Table 3.

Meta-Analysis results suggested that the overall prevalence is 0.06 (C.I. 0.03–0.10). Study heterogeneity was I^2^ = 86%; consequently, we attempted to group the studies according to “health status of dog” and heterogeneity lowered to I^2^ = 50% (*p* < 0.01). All details regarding the meta-analysis results has been showed in Figure 3. 

### 3.3. Klebsiella pneumoniae

The nine articles relating to the presence of *K. pneumoniae* in the dogs and considered for this study were published from 2002 [99] to 2019 [92]. The included papers reported a number of dogs that ranged from 50 [94] to 315 [92], for a total of 1448. In the included papers in this study, dogs were sampled mainly from rectal swabs [92,95,96,97] and urinary samples [98], while others considered fecal [93], blood [94], urine and fecal [34] samples, and one study from intravenous catheters [99]. Only one study [95] provided no info regarding the dog category while the other eight papers stated that sampling had been conducted on owned dogs. Regarding the health status of the sampled dogs, in one study they were healthy subjects [92], in four studies they were affected by urinary and intestinal infections [34] or urinary infections [98,99], or they have hospitalized in the Intensive Care Unit for other diseases [94], in two studies were not declared [92,96], and in two other studies both sick and healthy dogs [95,97] were sampled. In the Table 4, the information about the presence of *K. pneumoniae* in the dogs by the included studies is shown. In addition, Cetin and colleagues [98] conducted a multi-bacterial study also evaluating the presence of *P. aeruginosa* and *Enterobacter* spp.; also Sharif and colleagues [95] conducted a similar study, also assessing the presence of *Enterobacter* spp.; as well as Chanchaithong and colleagues [94] carried out a study which included the evaluation of the presence of *P. aeruginosa*; finally, Lobetti and coll. [99] performed their epidemiological study also on the presence of *Enterobacter* spp.

Meta-Analysis results suggested that the overall prevalence is 0.07 (C.I. 0.04–0.17). Study heterogeneity was I^2^ = 90%; consequently, we attempted to group the studies according to “health status of dog” and heterogeneity lowered to I^2^ = 78% (*p* < 0.01). All details regarding the meta-analysis results has been showed in Figure 4. 

### 3.4. Acinetobacter baumannii

Only four papers were included in our study. Regarding the type of samples, two articles [102,103] carried out rectal and oral swabs, while in one stool samples were taken [100] and in another skin swabs [101]. The number of positive dogs was less than 10 in all studies. Their percentages ranged from a minimum of 2.85% (3/205) [100] to a maximum of 8.82% (9/102) [102]; instead, the values shown by Mitchell and colleagues [101] and from Belmonte and colleagues [103] were 5.00% and 5.07%, respectively. All studies were conducted on owned dogs. Meanwhile, regarding their health status, two studies included hospitalized subjects for consultation or hospitalization [102,103], one study healthy subjects [101], another study healthy and hospitalized subjects [100]. The articles included in our study regarding the presence of this bacterium in dogs were published from 2014 to 2018. For other details, see Table 5.

Meta-Analysis results suggested that the overall prevalence is 0.04 (C.I. 0.02–0.08). Study heterogeneity was I^2^ = 23% (*p* = 0.25). All details regarding the meta-analysis results is shown in Figure 5. 

### 3.5. Pseudomonas aeruginosa

The studies included in this Review-Metanalysis were published from 2003 to 2018. The samples taken in the respective papers were heterogeneous as regards the sampling site (i.e., ocular and ear swabs) and type (i.e., blood, soft tissue, urine samples). The positivity rate ranged from 2% of urinary specimens in the study by Cetin and colleagues [98] and 31.62% of the ear samples in Penna and colleagues [107]. As shown in Table 5, the percentages of *P. aeruginosa* positivity in the other included papers ranged from 5.13 to 20.8% [46,82,94,104,105,106]. The article with the highest number of dogs sampled (*n* = 1182) was made by Ludwig and colleagues [104] who carried out an epidemiology survey involving different European countries. Only one study [107] provided no info regarding the dog category while the other seven papers stated that sampling had been conducted on owned dogs. In addition, all dogs included in papers about the presence of *P. aeruginosa* were sick of severe corneal ulcers [82], otitis externa, pyoderma and wounds [46], otitis externa [106,107], urinary tract infections [98], hospitalized with surgical, urinary, skin, and ear infections [100,101], and inpatient in Intensive Care Unit [94].

Meta-Analysis results suggested that the overall prevalence is 0.12 (C.I. 0.07–0.19). Study heterogeneity was I^2^ = 96%; consequently, we attempted to group the studies according to “health status of dog” and heterogeneity lowered to I^2^ = 37.6% (*p* < 0.01). All details regarding the meta-analysis results is shown Figure 6.

### 3.6. Enterobacter spp.

Six articles were considered eligible for inclusion in this study. All papers were published from 2002 to 2018. The number of dogs sampled ranged between 20 [108] and 136 [95], for a total of 450 dogs in the seven papers included. The prevalent type of specimens was represented by urine samples [98] and ocular tissues [108], and then by ocular, rectal, and oral swabs [82,95,109]. In addition, Lobetti and colleagues [99] has been performed their study on intravenous catheters. As shown in Table 7, the percentages of positivity ranged between 1.0 [99] and 21.3% [95]. Percentage values of *Enterobacter* spp. in dog urine were 2.0% [98]; in eye swabs they ranged from 4.2% [82] to 16.6% [108], while in oral swabs the values were 1.6% [109]; finally, in rectal swabs the values equal to 21.3% [95]. All studies included for the evaluation of the presence of *Enterobacter* spp. have been carried out on owned dogs, except the study by Sharif and colleagues [95] which provided no information in this regard. Of all the papers included, four were conducted on dogs with eye diseases [82,108] and urinary tract infections [98,99], one on healthy dogs [109], and another on both healthy and sick dogs [95]. 

Meta-Analysis results suggested that the overall prevalence is 0.30 (C.I. 0.17−0.44). Study heterogeneity was I^2^ = 50% (*p* = 0.11). All details regarding the meta-analysis results is shown in Figure 7.

## 4. Discussion

Our work aimed to systematically review data on the presence of pathogens bacteria ESKAPE in dogs, within the period of 2000 to 2019, focusing on the presence, percentage estimates, type of samples such as swabs and biological samples (i.e., feces, urines), dog category, and geographic distribution of the study (country and continent). The choice of the factors mentioned above and reported in the respective tables is related to the possibility of zoonotic transmission of the bacteria of the ESKAPE group from the dog to the human during AAT and AAA in health context. In particular, considering the great recent attention to these bacteria causing of nosocomial death by their characteristics of antibiotic resistance [110], the articles indexed and published in the last twenty years have been considered; the type of sample performed from the dog was highlighted to assess not only the variety of tropism of these bacteria but also to make a prediction of the body regions (of the dog) at risk of contamination and with which the patients/users involved make contact directly or indirectly; in our opinion, it was also noteworthy to consider the category of belonging of the dog (owned and non-owned) since the dogs that are involved in the AAIs, and in particular in the AAT, are owned dogs [5,6,111,112]. Finally, we also considered it appropriate to consider the geographical distribution of the studies obtained in order to have an overview of the interest of researchers in this topic and considering that AAIs are now widespread all over the world, especially in industrialized countries [113,114,115].

Our online search returned 52 papers published in different countries of the five Continents considered later of appropriate quality to provide useful data to evaluate the presence of these bacteria in the dogs. The geographical distribution of studies regarding ESKAPE bacteria group in the dog, in the present study is shown in the respective table to each bacterium.

As previously reported, we have considered the papers related to the presence of these bacteria in dogs since there is a large correlated lack in the control of dogs involved in AAT. In fact, this type of non-pharmacological therapy is carried out in settings such as hospitals or healthcare facilities and is often aimed at patients belonging to risk categories (e.g., dialysis patients, hospitalized patients, and immunosuppressed or immunocompromised patients) [5,116].

Despite the proven risk of these nosocomial opportunistic bacteria and the numerous investigations carried out both in the medical and veterinary fields, the studies concerning the dog involved in AAT and AAA in health context are very small. Particularly, the bacteria belonging to the MRSA group, together with *E. faecium*, represent the most studied bacteria of the ESKAPE group in the dog and mentioned in the *Guidelines of the American Journal of Infection Control* defined by Lefebvre and colleagues in 2008 [19], as well as in other scientific contributions published in international and national journals [117,118,119,120]. The lack of standardized control programs of ESKAPE in the dogs involved in AAT at the international level and in worldwide introduces a knowledge gap and makes it difficult to estimate related risk level for humans and thoroughly investigate transmission potential dynamics of these pathogens.

### 4.1. Enterococcus faecium

As reported by Bang and colleagues [27] Enterococci can be transmitted to humans via direct contact with animals. Moreover, as reported by Cinquepalmi and colleagues [70], the contact between pets and their owners is closer than in the past, therefore, contamination of urban roads with dog feces containing multidrug-resistant microorganisms is also a problem for public and environmental health. In our study, a total of 16 studies have been returned on the *E. faecium* presence in dogs but none have been carried out regarding the dogs involved in the AAIs or particularly in AAT and AAA in health context. Furthermore, as reported by Lefebvre and colleagues [19], dogs involved in AAIs should undergo health checks for resistant *E. faecium* as they can play a role in the spread of this nosocomial pathogen. The first report regarding the presence of *E. faecium* vancomycin resistant in dogs was made in 2002 by Simjee and colleagues [75] and the studies of other researchers have resumed with greater intensity since 2014. Recently, some studies were performed to highlight the transmission of *E. faecium* ampicillin- and vancomycin-resistant from the dog to humans [27,28,29,30,73]. Differently, Rodrigues and colleagues [74] as well as Kataoka and colleagues [68] showed that the strains of *E. faecium* isolated in their studies were resistant to many antibiotics but not to vancomycin, indicating that the risk of transmission of these strains to humans or the transfer of their resistance genes to others is limited bacteria belonging to the endogenous flora of the human. Almost all the studies included in this Review-Metanalysis isolated bacteria from feces [27,28,29,30] or rectal [65,66], oral [67], and different swabs were performed in remaining studies [69,72]. Interestingly, more recent studies have reconfirmed the dog’s status as a carrier of *E. faecium* but have also demonstrated its horizontal transfer to humans through contact and licking. Healthy livestock and pets can harbor enterococcal pathogens that can be transferred through the food chain, as well as through close associations such as embracing and licking humans [66]. Previously, the study by Jackson and colleagues [72] performed rectal and skin swabs, signaling the presence of *E. faecium* not only in the former but also in the neighboring regions of the abdomen and rear train. Furthermore, dogs (and cats) can act as reserves of antimicrobial resistance genes that can be transferred from pets to people [72]. Multi-drug resistant *E. faecium* has been isolated and identified from dogs with urinary tract infections supporting the hypothesis that enterococci are a true uropathogen and not just an opportunistic organism [64]. Finally, dogs are frequent carriers of CC17-related lineages and can play a role in the spread of this nosocomial pathogen [71].

Finally, based on the different clinical conditions of the dogs sampled in the papers included in this paper, also from the results of the Metanalysis it emerges that there is a greater prevalence of studies involving healthy [27,30,66,71,72,73] and “healthy and sick” dogs [29,64,65]. Therefore, in line with what reported by the American Guidelines [19], it should be mandatory to perform the microbiological control of the dog in the AAT and in the AAA in health context, avoiding the possible risk of transmission of *E. faecium*.

### 4.2. Staphylococcus aureus

Animal-to-human MRSA infection appears to be more evident in immunocompromised patients [17]. The bacteria belonging to the MRSA group, together with *E. faecium*, represent the bacteria of the ESKAPE group most studied in the dog and mentioned in the already mentioned Guidelines of the *American Journal of Infection Control* defined by Lefebvre and colleagues [19], but also in other scientific contributions published in international and national journals. While very few papers emphasize the need to monitor the presence of this bacterium in dogs involved in AAIs [117,118,119,120]. In line with the elective tropism of the staphylococci for the mucous membranes of the upper respiratory tract and for the skin, six studies were conducted on nasal swabs, six on nasal and perineal swabs, two on eye lesions, skin and pyogenic lesions, and one on ear swabs. It is interesting to note that inter-species transmission has been evaluated in all included studies not only from dog to human but also between the dog, farm animals and humans. In this regard, it is very interesting to underline the study by Lo Pinto and colleagues [84], in which the authors highlight the zoonotic risk for dogs of having a four times higher probability of contracting staphylococcal keratitis if belonging to people employed in veterinary work. Moreover, Hoet and colleagues [86] suggest that the owner’s profession was significantly associated, and the dogs owned by veterinarian students had a 20.5-fold probability (95% CI 4.5−93.6; *p*-value = 0.01) more likely to be MRSA positive compared to dogs owned by customers with different professions. The 2018 studies of Yadav and colleagues [79], Rahman and colleagues [80], and Kaspar and colleagues [81] highlighted the potential risk of transmission of MRSA strains from farm animals to dogs, highlighting how, again, the latter act as healthy and asymptomatic carriers of MRSA strains. In addition, the studies by Morris and colleagues [87] and Faires and colleagues [89] in which the possibility of colonization of dogs when sharing the domestic space with people or other animals with MRSA infection is evaluated.

Tabatabaei and colleagues [76] showed that pets and veterinarians could be potential sources of multidrug-resistant methicillin *S. aureus* (and multidrug-resistant methicillin *S. pseudintermedius*) in Iran. Ma and colleagues [77] carried out an epidemiological survey on dogs (and cats) showing a prevalence of MRSA equal to 2.6%. Dogs had a statistically more significant probability of carrying positive coagulase staphylococci than cats (*p* < 0.001). This study highlights important differences in the diversity and transport patterns of commensal staphylococci between dogs and cats in Australia. Huang and colleagues [78] carried out a study on the prevalence and characteristics of MRSA isolated from animals in shelter in Taiwan. The MRSA strains isolated in this study were like those already isolated from the human population in the past, indicating potential public health risks. Yadav and colleagues [79] carried out their study in India, evaluating the presence of MRSA in different animal species (cattle, buffalo, and dogs). In their article, eight strains of MRSA that exhibited methicillin resistance and possessed the *mec*A gene were isolated from dogs. It is interesting to note that the study by Kaspar and colleagues [81] highlighted the presence in the dog of the MRSA lineages typically described for cattle, underlining the impact of the spread of multi-drug resistant microorganisms. Drougka and colleagues [83], through the comparison between genetic markers, have shown that identical or very similar strains of MRSA spread between animals and veterinary staff. Pets harbor Panton-Valentine leukocidin (PVL) positive clones which are a possible source for transmission to humans. Tarazi and colleagues [85] found a strong association between the isolated strains of MRSA from dogs and those of humans that are in close contact with them (breeding centers and associated staff).

Pets can host pandemic strains of MRSA while residing in a family with an infected person. However, the source of MRSA for the animal cannot always be attributed to the human patient [87]. The presence of MRSA in apparently healthy and/or sick dogs makes it an emerging veterinary pathogen that could be considered a public health burden if widespread in our community outside hospitals [88]. Faires and colleagues [89] underlined the high prevalence of concomitant colonization of MRSA, and the identification of indistinguishable strains in humans and dogs (and cats) of the same family, suggesting the possible inter-species transmission of MRSA. Although the prevalence of colonization of these methicillin-resistant strains (*S. pseudintermedius*, *S. aureus* and *S. schleiferi* subsp. *coagulans*) was low, the combined prevalence of 3.1% can be a source of concern for both animal health and the public one [90].

At last, based on the health status of the dogs sampled in the papers included in our work and also from the results of the Metanalysis, it emerges that there is a greater prevalence of studies involving healthy [76,78,80,83,85,87,89] and “healthy and sick” dogs [81,86,88]. Therefore, consistently with what reported by the *American Guidelines* [19] and by the scientific contribution of other Authors [15,17,20], it should be mandatory to perform the microbiological control of the dog involved in the AAT and in the AAA in health context, to avoid the possible risk of transmission of these bacteria.

### 4.3. Klebsiella pneumoniae

Recently, the World Health Organization (WHO) published a global priority list of antibiotic-resistant bacteria and *K. pneumoniae* have been included in the “Priority 1: Critical” group (third-generation cephalosporin (3GC)-and/ or carbapenem-resistant Enterobacteriaceae) [26]. Marques and colleagues [36] reported the fecal colonization and sharing of *K. pneumoniae* clonal lineages between healthy humans and dogs living in close contact, suggesting the role of dogs as reservoirs of this bacterium, even though those strains were neither multidrug resistant nor hypervirulent.

The analyzed studies are uniformly distributed with regards to the sampling between rectal swabs, urinary and fecal samples, in line with the known enteric and urinary *K. pneumoniae* tropism, particularly in dogs.

The results of the study by Hong and colleagues [92] indicate the transmission and direct spread of extended-spectrum cephalosporin (ESC)-resistant Enterobacteriaceae, such as *K. pneumoniae*, between humans and pets. Zhang and colleagues [93] support the hypothesis of transfer of resistant bacteria between man and dog, since CTX-M-1 has been frequently found in fecal dog Enterobacteriaceae, while it is still rare in human Enterobacteriaceae in Canada, therefore suggesting the transfer of bacteria resistant to dogs from farm animals or other sources. Chanchaithong and colleagues [94] showed the high percentage of drug resistance among *K. pneumoniae* isolates, underlining that in the clinical practice of small animals’ routine detection of Extended-Spectrum Beta-Lactamase (ESBL)-producing bacteria is necessary, using reliable laboratory methods. The data shown by Liu and colleagues [34] highlight the alarming resistance to beta-lactamase in *Klebsiella* (and *Enterobacter*) species of canine origin in India, emphasizing them as indicators of antimicrobial resistance. Sharif and colleagues [95] emphasize the need for active surveillance studies on pets that live very close to humans, since inter-species transmission can occur within the same family. Gonzales-Torralba and colleagues [96] reported the first report concerning the isolation of bacteria that produce OXA-48 from pets. In particular, the clonal nature of *K. pneumoniae* would suggest nosocomial diffusion rather than repeated introduction by individual patients into the clinic. Abdel-Moein and colleagues [97] highlight the risk of transmission of *K. pneumoniae* infections, via the oro-fecal route, which could occur after handling infected pets or using contaminated objects within families.

From the data of the papers included in this work and from the results of the Meta-analysis, if on the one hand it emerged that the sampled dogs were mainly sick subjects [34,94,98,99]; on the other hand in two studies [95,97], both “healthy and sick” dogs were sampled. Instead, Zhang and colleagues [93] showed the presence of *K. pneumoniae* in healthy dogs. These findings additionally elucidate the need to perform the microbiological control of the dogs involved in the AAT and in the AAA in health context, avoiding the possible risk of transmission of this bacterium and contamination even in the home context.

### 4.4. Acinetobacter baumannii

Animals represent a potential reservoir of *A. baumannii* and can contribute to the dissemination of new emerging carbapenemases. The risk of transmission could increase in companion animals which are in more direct contact and closer vicinity with humans and are more prone to transfer or acquire *A. baumannii* [43]. In addition, *A. baumanni* has also been identified on the skin and in feces of healthy dogs [100,101]. The studies concerning the presence of *A. baumannii* denote the fairly recent attention towards this species/bacterial group (from 2014 to 2018) and despite the elective tropism of *A. baumanni* for the digestive system, heterogeneity of the samples taken (feces, rectal, oral, and skin swabs) was reported. Particularly, to be noted as all the studies indicated above were carried out in the context of veterinary hospitals or veterinary clinics. More specifically, Gentilini and colleagues [100] estimated the risk of colonization by Gram-negative, non-fermenting carbapenem-resistant bacteria in pets admitted to veterinary tertiary care centers, highlighting their potential role in the diffusion of resistance genes between animals and humans. The study reported by Mitchell and colleagues [100] showed that *Acinetobacter* spp. they can survive on the skin of dogs, which can become potential reservoirs of infection. Pailhoriès and coll. [102] have shown that *A. baumannii* strains are present in non-hospital settings on Reunion Island but Previously, Belmonte and colleagues [103] had assessed already the presence of *A. baumannii* in pets (dogs and cats) hospitalized in nine veterinary clinics on Reunion Island.

There are very few articles found in the literature and included in our work, therefore it is not scientifically correct to make conclusive statements on the risk of the transmission of *A. baumannii* from dog to man. Considering this, we can only emphasize that the dogs sampled in the study of Mitchell et at. [101] were healthy and that skin swabs had been performed, emphasizing the need to carefully monitor the dogs involved in AAT and AAA in the health context, considering the frequent contact that occurs in the respective settings.

### 4.5. Pseudomonas aeruginosa

In animals, particularly in dogs and cats, *P. aeruginosa* causes otitis external/media, corneal ulcers, urinary tract infection, and pyoderma [82,98,104]. However, only a few studies have highlighted the potential risk from contact with dogs infected with *P. aeruginosa*. In this regard, only Lefebvre and colleagues [20] have evaluated the presence of this bacterium in dogs visiting hospitalized people, while very few other studies have considered human–dog–environment transmission in the veterinary facilities, such as hospital and clinics [46,47,82].

The studies included in this Review-Metanalysis start from 2003 by Cetin and colleagues [98] (these authors conducted a multi-bacterial study also assessing the presence of *K. pneumoniae* and *Enterobacter* spp.) to arrive at 2018 with the paper by Ekapopphan and colleagues [82], (these authors also conducted a multi-bacterial study also evaluating the presence of MRSA and *Enterobacter* spp.). None of the studies included evaluated the presence of resistant *P. aeruginosa* in dogs involved in AAI. For the purposes of our investigation, highlighting the risk resulting from contact with certain body regions, the data reported by Penna and colleagues [107], which indicates the presence of *P. aeruginosa* in cases of canine external otitis and the data reported by Ludwig and colleagues [104] which indicates the presence of the bacterium in soft tissues, skin, and superficial wounds. As observed in human hospitals, *P. aeruginosa* acts as one of the multi-resistant microorganisms of veterinary clinical relevance [105]. Note the study by Lin and colleagues [46] on the antimicrobial resistance profiles of *P. aeruginosa* strains of canine origin in China, as it represents the first report of the oxacillin *bla*-OXA-31 resistance gene of this bacterium in a canine isolate.

From the data that emerged in our study, it appears clearly that *P. aeruginosa* causes urinary and auricular symptoms and infects soft tissues. In fact, all the included studies sampled sick or hospitalized dogs. Therefore, the risk of transmission to humans is mainly due to contact with symptomatic dogs but this does not exclude that there may be asymptomatic or reservoir dogs of this bacterium as a commensal of the urinary tract or ear.

### 4.6. Enterobacter spp.

*Enterobacter* spp. have been observed in device-related intravascular infections and surgical site infections (mainly postoperative or related to devices such as biliary stents). Transmission occurs through direct or indirect contact of the mucosal surfaces with the infectious agent (e.g., transfer from contaminated hands to contaminated neonatal units or urinals) or, in the case of endogenous flora, through transfer to adjacent body sites sensitive and sterile [48,49,50].

Few papers are present in the literature reporting epidemiological investigation data on the presence of *Enterobacter* spp. in dogs, and they considered its presence mainly in urine samples [94], intravenous catheters [99], ocular [82,108], oral [109], and rectal [95] swabs.

In 2003, Cetin and colleagues [98] were the first to report the presence of *Enterobacter* spp., in association with other bacterial species, in dogs with urinary problems. These authors conducted a multi-bacterial study evaluating the presence of *K. pneumoniae* and *P. aeruginosa*. The attention, after about 10 years of silence, was then directed to other body regions such as the oral cavity [109], the rectum [95], to get to the most recent studies of Ekapopphan and colleagues [82] and Lacerda and colleagues [108], which reported the presence of *Enterobacter* spp. in ocular swabs (cornea and conjunctiva). The alarming incidence of beta-lactamase resistance found in this study by Sharif and colleagues [95], could probably be the result of the indiscriminate use of antibiotics in veterinary practice, which reflects the possible risk of therapeutic failures that can occur in the treatment of infections caused by *Enterobacter* spp. Awoyomi and colleagues [109] highlighted that the oral cavities of hunting dogs can carry multi-resistant bacteria, of significant importance for public health since they could be transferred to humans through contaminated hunting tools and bite wounds.

The few studies included in this Review-Metanalysis underline the need not to underestimate the potential transmission risk of bacteria belonging to the *Enterobacter* genus, since based on the antibiotic-resistance data that emerged and the genetic analyzes carried out, these bacteria can be transferred through the bite wounds and more generally through contact with the dog’s external mucous membranes.

Finally, from the results of the Metanalysis, the sampled dogs were mostly sick subjects [82,98,99,108] but Sharif et al. [95] and Awoyomi et al. [109] showed positivity results in “healthy and sick” dogs and in healthy dogs, respectively. It is interesting to note that all the subjects sampled in the six studies concerning *Enterobacter* spp. included in this Review and Meta-analysis were dogs owned. Our results, also in this case, require reflection on the need to subject the dogs involved in the AAT and in the AAA in health context.

### 4.7. Potential Limitations of the Data

The notable lack of papers on the presence of ESKAPE bacteria in the dog involved in AAT at the international levels, as well as individual surveys poor representative, might have affected the true estimates of these zoonoses in individual countries, in the continents and across the world. This study did not utilized data from abstracts, posters, and conference proceedings but only full paper publications. Furthermore, the included studies showed a great heterogeneity regarding the number of dogs sampled and the respective percentages of positivity, as well as the samples taken; the isolation and identification methods of the individual bacteria were not considered; except for *Staphylococcus aureus* methicillin resistant, the inclusion of papers was made considering the resistance to antibiotics and not to specific antibiotic categories (ampicillin-, vancomycin-, β-lactam resistant).

## 5. Conclusions

Currently, there is moderate attention by researchers to the role of the dog as a vector of the bacteria of the ESKAPE group, the international scientific literature is still not very sensitive and is scarcely aimed at assessing the risk related to the presence of these bacteria in the dogs involved in the AAT and AAA in the health context. From the literature included in this work it emerges that the risk of dog–human zoonotic transmission (and vice versa) concerns all bacteria of the ESKAPE group, therefore the level of surveillance must include mandatory microbiological controls and strong rules of hygienic and behavioral management of the animal. In this regard, the continuous and constant health control of the animals involved in the AAIs, with particular reference to the dog, should be a priority. In fact, the close contact between human and pets determines the risk of zoonosis and creates opportunities for interspecies transmission of resistant bacteria [121].

In this aim, in our opinion, it is more and more necessary to a One Health approach, which involves the collaboration between veterinarians, physicians, public health operators, and epidemiologists, in order to prevent the transmission of such bacteria and to attain optimal health for humans, animals, and the environment. The AAIs, particularly the AAT, and the AAA in the health context, represent a concrete example of One Health approach and require necessarily an inter-disciplinary approach as they involve different health professionals. These operators, each according to their skills, work in team for people’s health, for the control and protection also of the health and welfare of the animal involved, for the prevention and control of zoonotic diseases.

Furthermore, we hope to be able to encourage a discussion with international experts regarding the need to draw up standardized hygiene-health-behavioral monitoring protocols, aimed at producing health and behavioral certifications valid for all dogs that perform AAT and AAA in the health context.

## Figures and Tables

**Figure 1 ijerph-17-03278-f001:**
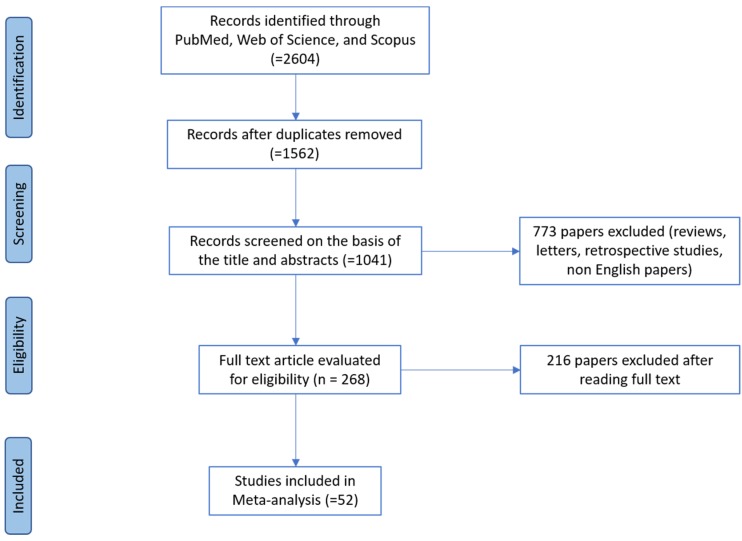
PRISMA (Preferred Reporting Items for Systematic reviews and Meta-Analyses) flow diagram.

**Figure 2 ijerph-17-03278-f002:**
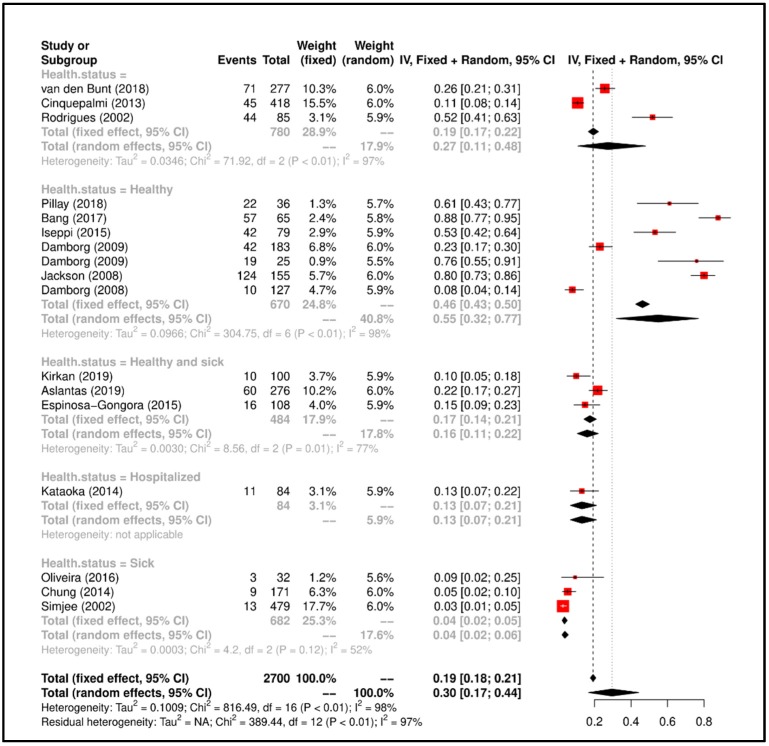
Forrest plot corresponding to occurrences of the *E. faecium* grouped by “health status of dog”.

**Figure 3 ijerph-17-03278-f003:**
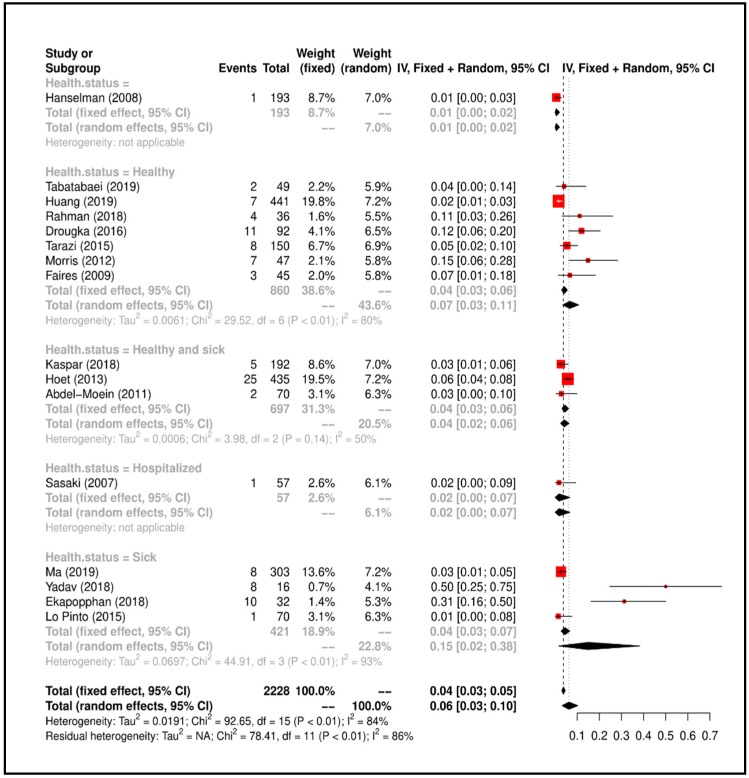
Forrest plot corresponding to the occurrences of *S. aureus* grouped by “health status of dog”.

**Figure 4 ijerph-17-03278-f004:**
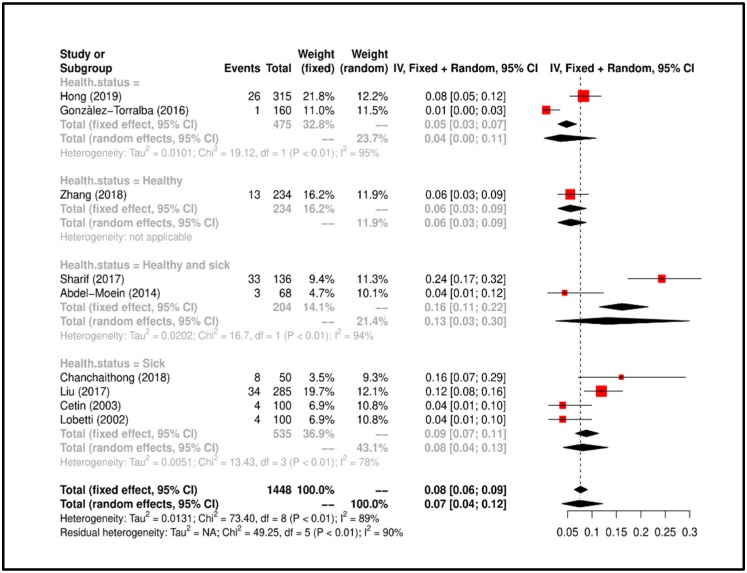
Forrest plot corresponding to occurrences of *K. pneumoniae* grouped by “health status of dog”.

**Figure 5 ijerph-17-03278-f005:**
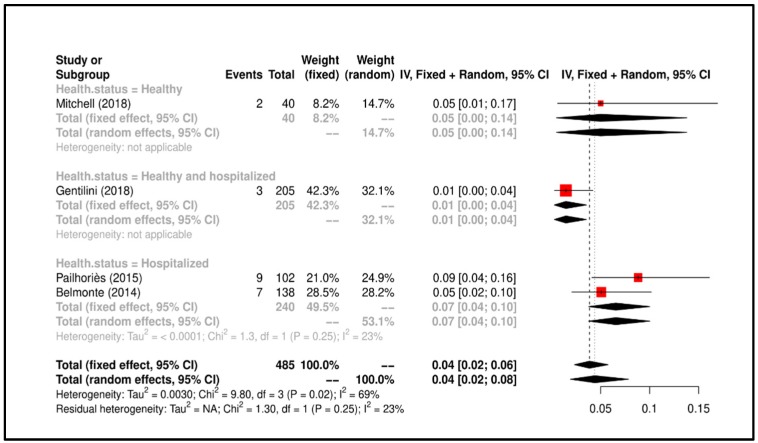
Forrest plot corresponding to occurrences of *A. baumannii* grouped by “health status of dog”.

**Figure 6 ijerph-17-03278-f006:**
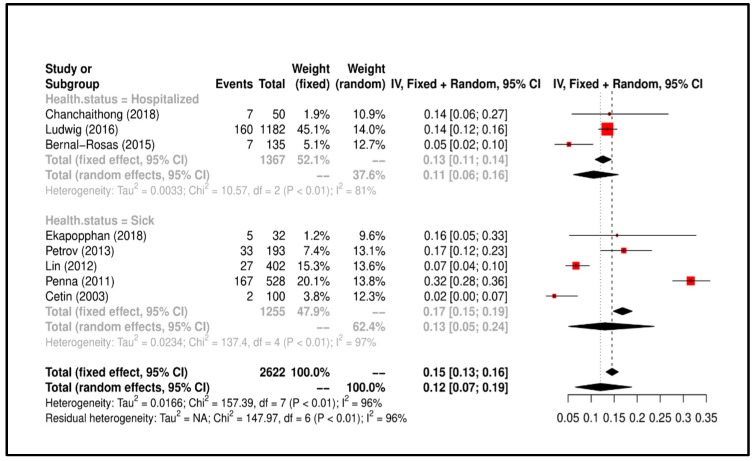
Forrest plot corresponding to occurrences of *P. aeruginosa* grouped by “health status of dog”.

**Figure 7 ijerph-17-03278-f007:**
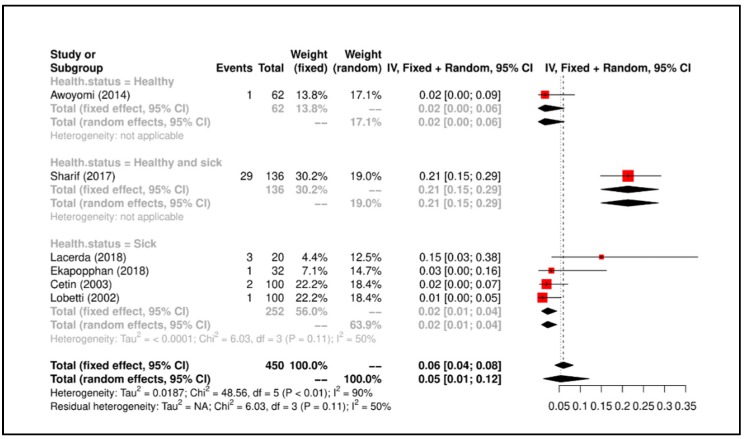
Forrest plot corresponding to occurrences of *Enterobacter* spp. grouped by “health status of dog”.

**Table 1 ijerph-17-03278-t001:** Number of studies of most relevant pathogens in the different geographical areas.

Bacterial Species	Considered Continent					Subtotal of Studies
	Africa	Americas	Asia	Europe	Oceania	
*Enterococcus faecium*	1	2	3	10	-	16
*Staphylococcus aureus*	1	5	7	2	1	16
*Klebsiella pneumoniae*	2	1	4	2	-	9
*Acinetobacter baumannii*	-	-	-	4 *	-	4
*Pseudomonas aeruginosa*	-	2	3	3	-	8
*Enterobacter* spp.	2	1	2	1	-	6
Total of included Studies						52 *

* The total number of studies was obtained by considering multi-bacterial studies only once [82,94,95,98,99].

**Table 2 ijerph-17-03278-t002:** Data of the included articles regarding the presence of *Enterococcus faecium* dogs.

First Author, Year of Publication	No. of Sampled Dogs	Type of Samples	Dog Category	Health Status of Dog	No. of Positives	%	Reference
Kirkan, 2019	100	urines	Owned	Healthy and sick	22	22	[64]
Aslantas, 2019	276	rectal swabs	Owned	Healthy and sick	60	21.7	[65]
van den Bunt, 2018	277	feces	Owned	Not Available	71	25.6	[28]
Pillay, 2018	36	rectal swabs	Owned	Healthy	22	12.0	[66]
Bang, 2017	65	feces	Military working	Healthy	57	87.7	[27]
Oliveira, 2016	32	oral swabs	Owned	Sick	3	9.4	[67]
Iseppi, 2015	79	feces	Owned	Healthy	42	36.5	[30]
Espinosa-Gongora, 2015	108	feces	Owned	Healthy and sick	16	14.9	[29]
Kataoka, 2014	84	feces	Owned	Hospitalized	11	15.7	[68]
Chung, 2014	171	swabs	Owned	Sick	9	5.3	[69]
Cinquepalmi, 2013	418	feces	N.A.	Not Available	45	10.76	[70]
Damborg, 2009	183	feces	Owned	Healthy	42	23.0	[71]
	25				19	76.0	
Jackson, 2008	155	swabs	Owned	Healthy	124	80.0	[72]
Damborg, 2008	127	feces	Owned	Healthy	10	8.0	[73]
Rodrigues, 2002	104	feces	Not available	Not available	44	42.3	[74]
Simjee, 2002	479	urines	Owned	Sick	13	2.71	[75]

**Table 3 ijerph-17-03278-t003:** Data of the included articles regarding the presence of *Staphylococcus aureus* in dogs.

First Author, Year of Publication	No. of Sampled Dogs	Type of Samples	Dog Category	Health Status of Dog	No. of Positives	%	Reference
Tabatabaei, 2019	49	nasal and perineum swabs	Owned	Healthy	2	4.01	[76]
Ma, 2019	303	nasal and perineum swabs	Owned	Sick	8	2.6	[77]
Huang, 2019	441	nasal swabs	Sheltered	Healthy	7	1.6	[78]
Yadav, 2018	16	pyogenic lesions	Owned	Sick	8	50.0	[79]
Rahman, 2018	36	nasal swabs	Owned	Healthy	4	13.8	[80]
Kaspar, 2018	192	nasal and perineum swabs	Owned	Healthy and Sick	5	2.6	[81]
Ekapopphan, 2018	32	ocular swabs	Owned	Sick	10	31.25	[82]
Drougka, 2016	92	nasal and perineum swabs	Not Available	Healthy	11	10.8	[83]
Lo Pinto, 2015	70	nasal swabs	Owned	Sick	1	1.42	[84]
Tarazi, 2015	150	nasal swabs	Owned, Stray and Farm	Healthy	8	12.7	[85]
Hoet, 2013	435	nasal and perineum swabs	Owned	Healthy and Sick	25	5.7	[86]
Morris, 2012	47	nasal and perineum swabs	Owned	Healthy	7	14.8	[87]
Abdel-Moein, 2011	70	swabs	Owned	Healthy and Sick	2	2.9	[88]
Faires, 2009	45	nasal swabs	Owned	Healthy	3	6.6	[89]
Hanselman, 2008	193	nasal and rectal swabs	Owned	Not Available	1	0,5	[90]
Sasaki, 2007	57	nasal swabs	Owned	Hospitalized	1	1.7	[91]

**Table 4 ijerph-17-03278-t004:** Data of the included articles regarding the presence of *Klebsiella pneumoniae* in dogs.

First Author, Year of Publication	No. of Sampled Dogs	Type of Samples	Dog Category	Health Status of Dog	No. of Positives	%	Reference
Hong, 2019	315	rectal swabs	Owned	Not Available	26	8.3	[92]
Zhang, 2018	234	feces	Owned	Healthy	13	5.55	[93]
Chanchaithong, 2018	50	blood	Owned	Sick	8	16.0	[94]
Liu, 2017	285	feces and urines	Owned	Sick	34	12.0	[34]
Sharif, 2017	136	rectal swabs	Not Available	Healthy and sick	33	24.2	[95]
Gonzàlez-Torralba, 2016	160	rectal swabs	Owned	Not Available	1	0.6	[96]
Abdel-Moein, 2014	68	rectal swabs	Owned	Healthy and Sick	3	2.7	[97]
Cetin, 2003	100	urines	Owned	Sick	4	4.0	[98]
Lobetti, 2002	100	intravenous catheters	Owned	Sick	4	4.0	[99]

**Table 5 ijerph-17-03278-t005:** Data of the included articles regarding the presence of *Acinetobacter baumannii* in dogs.

First Author, Year of Publication	No. of Sampled Dogs	Type of Samples	Dog Category	Health Status of Dog	No. of Positives	%	Reference
Gentilini, 2018	205	feces	Owned	Healthy and hospitalized	3	2.85	[100]
Mitchell, 2018	40	skin swab	Owned	Healthy	2	5.00	[101]
Pailhoriès, 2015	102	rectal and oral swabs	Owned	Hospitalized	9	8.82	[102]
Belmonte, 2014	138	rectal and mouth swabs	Owned	Hospitalized	7	5.07	[103]

**Table 6 ijerph-17-03278-t006:** Data of the included articles regarding the presence of *Pseudomonas aeruginosa* in dogs.

First Author, Year of Publication	No. of Sampled Dogs	Type of Samples	Dog Category	Health Status of Dog	No. of Positives	%	Reference
Ekapopphan, 2018	32	corneal and conjunctival swabs	Owned	Sick	5	20.8	[82]
Chanchaithong, 2018	50	blood	Owned	Hospitalized	7	14.0	[94]
Ludwig, 2016	1182	soft tissues	Owned	Hospitalized	160	13.5	[104]
Bernal-Rosas, 2015	135	clinical samples	Owned	Hospitalized	7	5.13	[105]
Petrov, 2013	193	ear swabs	Owned	Sick	33	17.0	[106]
Lin, 2012	402	soft tissues	Owned	Sick	27	6.7	[46]
Penna, 2011	528	ear swabs	Not Available	Sick	167	31.62	[107]
Cetin, 2003	100	urines	Owned	Sick	2	2.0	[98]

**Table 7 ijerph-17-03278-t007:** Data of the included articles regarding the presence of *Enterobacter* spp. in dogs.

First Author, Year of Publication	No. of Sampled Dogs	Type of Samples	Dog Category	Health Status of Dog	No. of Positives	%	Reference
Lacerda, 2018	20	conjunctival surface and aqueous humor	Owned	Sick	3	16.6	[108]
Ekapopphan, 2018	32	corneal and conjunctival swabs	Owned	Sick	1	4.2	[82]
Sharif, 2017	136	rectal swabs	Not Available	Healthy and sick	29	21.3	[95]
Awoyomi, 2014	62	oral swabs	Owned	Healthy	1	1.6	[109]
Cetin, 2003	100	urines	Owned	Sick	2	2.0	[98]
Lobetti, 2002	100	intravenous catheters	Owned	Sick	1	1.0	[99]

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
