# Peer review of "Systematic Review and Meta-Analysis of the Occurrence of ESKAPE Bacteria Group in Dogs, and the Related Zoonotic Risk in Animal-Assisted Therapy, and in Animal-Assisted Activity in the Health Context"

_ijerph, 2020, doi:10.3390/ijerph17093278_

Round 1
Reviewer 1 Report
Authors presented a importance analysis of Enterococcus faecium, Staphylococcus aureus, Klebsiella pneumoniae, Acinetobacter baumannii, Pseudomonas aeruginosa and Enterobacter spp at the context of assisting animals using in human activities.
Although the work is worthwhile, taking into account the work required to carry out a meta-analysis, I believe that in general it is very massive. It should be substantially shortened, mainly in the results section. I suggest to the authors presente a tables of results of “Most relevant pathogens in the different geographical areas”.
Transpose the main results and discussion to these sections.
Please review the use of virgules in all the manuscript. For example, lines 18 and 19“Particularly, animal‐assisted therapies, and animal‐assisted activities sometimes, are performed in 19 health facilities, hospitals, etc” and “136 process (number of dogs, percentage of positive dogs etc”
Line 19 : “health facilities, hospitals, etc.”. In fact, a hospital is a health facilities, so this sentence is redundant.
Keywords : Must be revised. The keywords are used to find the jobs that interest us when we do a database search. The acronym AAI is not sufficiently descriptive to constitute a keyword. Remove or write “Animal Assisted Interventions”. Use only “zoonoses” or “zoonotic risk” because are about the same. Besides it is widely acronym at this area, if we search in google for AAI, only on the 2nd page does the meaning appear. A good acronym is not necessarily a good keyword
Line 48 “activities [] such as petting”, mistake in the []
Line 60 “ESKAPE was used for the first time in 2008 by L.B. Rice (put the reference)” Mistake on parenthesis.
Line 66 “Enterococcus faecium is commensal microorganisms”, plural is wrong
Line 72 “soft tissue infections infective endocarditis”
Line 72 – It doesn’t make any sense mix bacteremia with the other diseases because bacteremia can occurs in any disease and it’s not a disease, but one of the stages of the pathogenic process.
Line 85 – bacteremia is again wrong used, remove.
Line 162 “16 articles 162 reporting on Enterococcus faecium [27‐30,60‐71]; 16 reporting on MRSA [72‐87]; nine reporting on 163 Klebsiella pneumoniae [34,88‐95]; four on Acinetobacter baumannii [96‐99]; eight on Pseudomonas 164 aeruginosa [45,78,90,94,100‐103]; and six studies reporting on Enterobacter spp.”
Please use the contracted or extensive form of the numbers, not the two forms.
-All the values of heterogeneity should be present with p value too. For example “heterogeneity lowered to I2 =52% (p < …)”
Line 202 “1 (0.5%) and 25 (5.7%); in the second, between 1 (1,425) and 8 (12.7%);” Present all the results in percentage
Please improve the definition of Forrest plot figures
In some cases, the species name are present in non-italic (I’m not talking about when are used as keywords in database search). Examples lines 237 to 240
Line 268 “ made in Americas (both conducted in South America)”, Why you don’t write just “made in South America”?
Lines 281 to 285 – “Furthermore, Cetin et al. [94] conducted a "multi‐bacterial" study also 282 evaluating the presence of K. pneumoniae and Enterobacter spp.; also Ekopopphan et al. [78] 283 conducted a similar study, also assessing the presence of MRSA and Enterobacter spp.; as well as 284 Chanchaithong et al. [90] carried out a study which included the evaluation of the presence of K. 285 pneumoniae.”
I don’t see any utility in the referencing in a block of authors without presenting results. Remove the entire paragraph.
Reviewer 2 Report
Very nice and important work.
Please check and unify the spelling of the latin names in italic.
It think it also would be better to use "author and coworkers" or "author and colleagues" than "author et al." in text citations.
detailed comments_
Line 461: maybe write out "WHO" in the first appearance
Line 474: the same for "ESC"
Line 476 "CTX-M-1" does not have to be in brackets
Line 481: write out "ESBL"
Line 494: restructe sentence "These findings too, ...." e.g. "These findings additionally elucidate the need ..."
line 536: shouldn´t it be "AAI"
line 563: maybe better "silence" than "emptiness"
Reviewer 3 Report
This review and meta analysis is welcome because it addresses an important but less focussed area towards understanding of One Health. My main concerns are however about lack of elaboration on the methodology part and conclusion.
Line 135 : please provide clarity on 'characteristics of the samples' and 'sampling process' that determined the exclusion/inclusion.
typo line 143: reported
Similarly in continuation section 2.4 is also needs clarity viz. 'predetermined' and ' 'methodological clear information'
Please provide reference for R ( not only for the package but also the programming environment)
The authors state in results that if pooling of the prevalence weights ended in high heterogeneity the studies were grouped according to health status. I could not understand this grouping. Please provide details- eg. which articles were removed to obtain homogeneity?
It is suggested to report only those results, that are part of the discussion. As it appears now the results are too many and difficult to relate with in the discussion.
In your tables you provide healthy and sick dog numbers only for few studies, why the data from others is not reflected. Further your figure 2 has 17 studies but you discuss only 16 through out for results on E.faecium.
The Discussion section can be considerably reduced.
The conclusion mentions One Health approach, but the approach has not been discussed enough though out the discussion. Please bring out how the collaborations between different stakeholders can be achieved.
Round 2
Reviewer 1 Report
I really enjoyed the reviewed work of the authors. In fact, the first table gives us a general idea of the manuscript' geographical approach. I suggest to the authors in subsequent works, try to use a world map with the number of works, just to be differentiate from this work.
The new tables and sentence editing made the work more fluid and the scientific inaccuracies, namely the species reference form, were, in my view, all mitigated.
I think that at this stage, the work should be accepted by the scientific community.